# Effects of Resveratrol on Growth Performance, Intestinal Development, and Antioxidant Status of Broilers under Heat Stress

**DOI:** 10.3390/ani11051427

**Published:** 2021-05-17

**Authors:** Chi Wang, Fei Zhao, Zhen Li, Xu Jin, Xingyong Chen, Zhaoyu Geng, Hong Hu, Cheng Zhang

**Affiliations:** 1Department of Animal Science, College of Animal Science and Technology, Anhui Agricultural University, Hefei 230036, China; nicksix6@126.com (C.W.); zhaofei550516@163.com (F.Z.); li1819559237@163.com (Z.L.); jinxu1228@163.com (X.J.); chenxingyong@ahau.edu.cn (X.C.); gzy@ahau.edu.cn (Z.G.); 2Department of Animal Science, College of Animal Science, Anhui Science and Technology University, Fengyang 233100, China; haiyanghh@163.com

**Keywords:** broiler, heat stress, resveratrol, intestinal characteristic, antioxidant status

## Abstract

**Simple Summary:**

Broilers have unique physiological characteristics, no sweat glands and full of feathers, which makes it difficult to dissipate heat in high-temperature environments and is prone to heat stress (HS). HS has strong adverse effects on the meat production, growth performance, intestinal morphology, mortality and welfare of broilers, which can be alleviated by nutrition regulation. Resveratrol has been found to reduce the damage of HS on meat quality, immune and inflammatory response of broilers. However, there are few reports on the effects of resveratrol on the intestinal development and antioxidant capacity of broilers under HS. We demonstrated that resveratrol could improve the intestinal development and growth performance of broilers under HS. Besides, these findings suggest that resveratrol may offer an effective nutritional strategy to improve intestinal antioxidant function by regulating the expression of critical factors in the Nrf2 signaling pathway.

**Abstract:**

The study investigated resveratrol’s effect on growth performance, intestinal development, and antioxidant capacity of broilers subjected to heat stress (HS). A total of 162 21-day-old male AA broilers were randomly divided into 3 treatment groups with 6 replicates of 9 birds each. The 3 treatment groups were as follows: the control (CON), in which broilers were housed at 22 ± 1 °C for 24 h day^−1^, and the HS and HS + resveratrol (400 mg/kg) groups, in which broilers were housed at 33 ± 1 °C for 10 h a day from 8:00 to 18:00 and 22 ± 1 °C for rest of the time. Results indicated that birds in the HS group exhibited lower (*p* < 0.05) final body weight (BW) and average daily gain (ADG) compared with birds in the CON group. HS birds also had lower (*p* < 0.05) relative jejunum weight, relative ileum and jejunum length, jejunal villus height, and villus height to crypt depth ratios than the CON group. The activities of glutathione peroxidase (GPX), glutathione S-transferase (GST), superoxide dismutase (SOD), and the mRNA levels of NF-E2-related factor 2 (*Nrf2*), *SOD1*, and *GPX* were also lower (*p* < 0.05) in the HS than CON group. The HS group had higher (*p* < 0.05) protein carbonyl (PC) contents and Kelch-like ECH-associated protein 1 (*Keap1*) mRNA levels. Compared with HS group, the HS + resveratrol group exhibited higher (*p* < 0.05) BW and ADG, relative jejunum weight, relative length of ileum, jejunal villus height, activities of GPX and GST, and mRNA levels of *Nrf2* and *SOD1*, but they had lower (*p* < 0.05) PC content and *Keap1* mRNA levels. In conclusion, resveratrol can improve the intestinal development and antioxidant function of broilers under HS, and therefore improve growth performance. The mechanism by which resveratrol enhances the intestinal antioxidant capacity is mediated by Nrf2 signaling pathway.

## 1. Introduction

Humans and animals maintain a dynamic balance between heat generation and dissipation. Heat stress (HS) will occur when the ambient temperature exceeds the thermal neutral zone of animals [1]. The excessive increase of reactive oxygen species (ROS) under HS stimulation leads to the destruction of redox homeostasis and ultimately causes oxidative damage to lipids, proteins, and DNA [2], adversely affecting animal health. As global warming intensifies, HS has been a critical factor restraining animal production. Due to high metabolic activity, lack of sweat glands, and insulating feathers, which lead to heat dissipation difficulties, broilers are more susceptible to HS [3]. HS adversely affects the physiology, behavior, immune response, food quality, and safety of poultry. Specifically, HS can induce intestinal morphology damage and dysfunction [4,5], high intestinal permeability, and disordered caecal microflora [5,6,7], and cause oxidative stress damage and immune injury [6,8]. These combined factors lead to increased mortality and substantial economic losses to the global poultry industry [9,10]. Therefore, exploring HS relief and its mechanism of action has become an important research focus.

Numerous studies have shown that nutrition is an effective means to regulate HS, such as vitamin E, vitamin C [11] and glutamine [12]. Resveratrol (RES) is a naturally occurring polyphenolic compound found in various plants, including grapes, Polygonum cuspidatum, and peanuts [5]. It has several biological effects, including anti-inflammation [13], antioxidation [14], and energy metabolism regulation [15]. Mayangsari and Suzuki [16] found that resveratrol could protect the integrity of human Caco-2 colonic epithelial cells’ tight junctions and improve intestinal epithelial barrier function. Zhao et al. [17] reported that resveratrol could maintain the intestinal barrier’s integrity and reduce intestinal damage by inhibiting the apoptosis of intestinal epithelial cells of rats. Our previous study found that resveratrol could improve the muscle antioxidant function of broilers reared under normal ambient temperature [18]. Moreover, our previous study showed that resveratrol exerted beneficial effects on intestinal morphology [5], the spleen, and muscle antioxidant capacity [3] of broilers under heat stress.

Resveratrol’s mechanism of action remains to be explored. Nuclear factor erythroid 2-related factor 2 (Nrf2) is an important transcription factor in the antioxidant system, which regulates organic antioxidant capacity [19,20] by influencing the expression of heme oxygenase-1 (*HO-1*), glutathione-S transferase (*GST*), NAD (P)H/quinone oxidoreductase 1 (*NQO1*), and glutamate-cysteine ligase. Previous reports pointed out that resveratrol could increase the expression of *Nrf2*, which activates the expression of its downstream antioxidant genes and scavenges free radicals to relieve oxidative stress damage. Hence, we speculated that resveratrol exerts a role in protecting intestinal health by regulating antioxidant genes related to the Nrf2 signaling pathway. However, there are few reports on the effects of resveratrol on the intestinal health and antioxidant capacity of broilers under HS. Thus, the current study investigated resveratrol’s effects on growth performance, intestinal development, and antioxidant status of broilers under HS.

## 2. Materials and Methods

### 2.1. Animals, Diets and Experimental Design

All experimental animal treatments were approved by the Animal Ethics Committee of Anhui Agricultural University, Hefei, China (approval code: 2020117). Diet was formulated according to our previous reports [21]. A total of 162 21-old-days Arbor Acres broilers with similar weight (the average weight was 700 ± 20 g) were divided into three groups: the control group (CON), HS group, and HS + resveratrol group. Each treatment contained 6 replicates of 9 broilers. The CON broilers were housed at thermal comfort conditions (22 ± 1 °C) for 24 h day^−1^, the birds of HS and HS + resveratrol groups were exposed to 33 ± 1 °C for 10 h a day from 8:00 to 18:00 during the 21-day experimental phase, during the remaining time temperature was consistent with the CON. The broilers of the CON group and the HS group were fed the basal diet (Table 1), the broilers of the HS + resveratrol group were fed the basal diet contained with 400 mg/kg resveratrol in accordance with previously described [1,5]. The temperature and the relative humidity were automatically controlled by the environmental control systems (Big Herdsman, Qingdao, China). The lighting program was 23 h light and 1 h dark per day, and the relative humidity was between 60–70% throughout the experimental period. The basal diet showed the basic composition and nutrients requirements.

### 2.2. Data Collection and Sampling

Final broiler body weights (BW) were measured at the beginning (21 days of age) and the end (42 days of age) of the trial. Feed intake was recorded daily. The average daily gain (ADG), average daily feed intake (ADFI), and feed to gain ratio (F/G) were calculated per bird and adjusted by mortality. At the end of the trial, two broilers close to average BW were selected from each replicate, weighed, and then euthanized by cervical dislocation. The duodenum (from the gizzard to the bile duct), jejunum (from the bile duct to Meckel’s diverticulum), and ileum (from Meckel’s diverticulum to the ileocecal junction) were isolated, and their relative length (cm kg^−1^ live BW) and relative weight were (g kg^−1^ live BW) measured. A 2 cm segment was taken from the middle part of the jejunum, washed in ice-cold physiological saline solution, and fixed in 10% formalin for morphology measurement. The remaining jejunum samples were washed in ice-cold physiological saline solution and then were used to scrape the mucosa, which was immediately snap-frozen in liquid nitrogen until analysis.

### 2.3. Jejunum Morphology Analysis

Jejunum morphology was measured as previously described [6,22]. The formalin-fixed jejunal samples were dehydrated, embedded, and stained with hematoxylin and eosin. Villus height and crypt depth were determined using an image processing and analysis system (Leica Imaging Systems Ltd., Cambridge, UK). Villus height to crypt depth ratio was calculated.

### 2.4. Antioxidant Activity in the Jejunum

The jejunum mucosa was homogenized in ice-cold 0.9% physiological saline solution (1:9 *w*/*v*), and the homogenate was centrifuged at 2500× *g* for 10 min at 4 °C, then the supernatant was used to antioxidant enzyme activity measurement at appropriate dilution. The supernatant’s protein content was assayed by the Bradford method with bovine serum albumin as the standard. Jejunum catalase (CAT), glutathione peroxidase (GPX), glutathione reductase (GR), glutathione S-transferase (GST), superoxide dismutase (SOD), malondialdehyde (MDA), and protein carbonyl (PC) levels were measured using corresponding commercial assay kits (Nanjing Jiancheng Institute of Bioengineering, Nanjing, China) according to the manufacturer’s instructions [1].

### 2.5. Real-Time PCR Analysis

Total RNA was isolated from jejunum mucosa using TRIzol TM Reagent (Thermo Fisher Scientific, Waltham, MA, USA), and then reverse-transcribed into cDNA using the Hifair^®^ II 1st Strand cDNA Synthesis SuperMix for qPCR (gDNA digester plus) (Yeasen, Shanghai, China) in accordance with the corresponding instructions. The mRNA expression levels of the genes were quantified using Real-Time PCR with a 7500 Software Real-Time PCR System (Thermo Fisher Scientific, Waltham, MA, USA) and Hieff^®^ qPCR SYBR Green Master Mix (Low Rox Plus) (Yeasen, Shanghai, China). The primers (Table 2) for *Nrf2*, kelch-like ECH-associated protein 1 (*Keap1*), *HO-1*, *NQO1*, *CAT*, *SOD1*, *GST*, and *GPX* were designed via Primer 5.0 software and commercially synthesized by Generalbiol Corporation (Anhui, China). 10 μL of Hieff^®^ qPCR SYBR Green Master Mix, 0.4 μL of each primer, 8.8 μL ddH2O and 0.4 μL of cDNA sample made up the PCR mixture. The PCR cycling parameters followed our previous study [1]. The β-actin gene was used as the internal reference gene, and the mRNA expression level of the target gene was calculated according to the 2^−ΔΔCt^ method.

### 2.6. Statistical Analysis

Data were analyzed using SPSS 18.0. Statistical differences among groups were determined via one-way ANOVA followed by Duncan’s multiple-range tests. Differences were considered significant at *p* < 0.05, and all data were presented as means with their standard errors.

## 3. Results

### 3.1. Growth Performance

Growth performance data are shown in Table 3. Birds in the HS group exhibited lower (*p* < 0.05) BW and ADG compared with CON group. Moreover, birds in the HS + resveratrol group exhibited higher (*p* < 0.05) BW and ADG than those in the HS group. There were no statistically significant differences between CON and HS + resveratrol groups.

### 3.2. Intestinal Characteristics

The relative weight and length of the duodenum, ileum and jejunum of broilers are presented in Table 4. The relative jejunum weight and the relative length of ileum and jejunum were decreased (*p* < 0.05) in the HS group compared to the CON group. The relative jejunum weight and the relative length of ileum were increased (*p* < 0.05) in HS + resveratrol group compared with those in the HS group. There were no statistically significant differences between CON and HS + resveratrol groups.

### 3.3. Intestinal Morphology

Data for jejunum morphology are shown in Table 5. Compared with birds in the CON group, birds in the HS group exhibited lower (*p* < 0.05) villus height and villus height to crypt depth ratio. Compared with birds in the HS group, birds in the HS + resveratrol group exhibited greater (*p* < 0.05) villus height. There were no statistically significant differences between CON and HS + resveratrol groups.

### 3.4. Antioxidant Status in the Jejunum

Data for intestinal antioxidant status are shown in Table 6. Broilers in the HS group exhibited lower (*p* < 0.05) GPX, GST, and SOD activity and higher (*p* < 0.05) PC activity than those in the CON group. Broilers in the HS + resveratrol group exhibited higher (*p* < 0.05) GPX and GST activity and lower (*p* < 0.05) PC content than those in the HS group. There were no statistically significant differences between CON and HS + resveratrol groups.

### 3.5. Nrf2 Signaling Pathway-Related Genes mRNA Expression Levels

The mRNA expression levels of the Nrf2 signaling pathway-related genes in the jejunum are presented in Table 7. Broilers in the HS group exhibited lower (*p* < 0.05) *Nrf2*, *SOD1*, and *GPX* mRNA levels and higher (*p* < 0.05) *Keap1* mRNA levels than those in the CON group. Broilers in the HS + resveratrol group exhibited higher (*p* < 0.05) *Nrf2* and *SOD1* mRNA levels and lower (*p* < 0.05) *Keap1* mRNA levels than those in the HS group. There were no statistically significant differences between CON and HS + resveratrol groups.

## 4. Discussion

HS is an important environmental factor that limits broiler production and can cause physiological disorders that lead to a decrease in broilers’ antioxidant and immune function [23,24]. A study by Liu et al. [14] showed that the ADG and ADFI of birds subjected to HS were significantly reduced. Similarly, the present research showed that HS significantly decreased BW and ADG of broilers. However, we haven’t found a significant difference in ADFI, which may be related to the lower heat-treated temperature (33 ± 1 °C) of the present study compared to the study of Liu et al. (37 ± 2 °C) [14]. The reasons may be due to an insufficient blood supply to the digestive systems of broilers under HS, which affects its growth and physiological functions. Multiple studies confirmed that the performance degradation of broilers caused by HS could be alleviated by nutritional regulation [14,25]. Liu et al. [14] have shown that the addition of 400 mg/kg resveratrol to the diet could effectively alleviate the adverse effects of HS on the growth performance of black-boned chickens, indicating the addition of resveratrol reduces the inflammatory response in the intestines and improves the utilization of nutrients in the intestines, improving the growth performance of broilers. Our present study also confirmed an increase in the growth performance of heat-stressed broilers, which were fed a diet with resveratrol. The above results strengthen the rationale for applying resveratrol to broiler production to improve broilers’ growth performance under HS.

It has been confirmed that intestinal architecture and development play a vital role in the digestion and absorption process of the gut and that intestinal health and function are usually reflected by villus height, crypt depth, and villus height to crypt depth ratio [21]. Several studies showed that HS led to increased crypt depth and decreased villus height and villus height to crypt depth ratio in broilers [5,6,8]. Similarly, this study indicated that HS was harmful to the morphology of the jejunum. The reason may be that HS reduced intestinal blood flow and caused intestinal ischemia. This ischemia can lead to epithelial shedding, which led to shortened villus height and deepening of crypts [26]. Also, the present study found that HS decreased the relative jejunum weight and the relative length of the ileum and jejunum. Therefore, HS is harmful to the intestinal development of broilers. Liu et al. [27] showed that resveratrol could significantly increase the jejunal villus height and the villus height to crypt depth ratio of broilers under HS, while the crypt depth was decreased. Zhang et al. [5] reported that resveratrol could alleviate HS’s adverse effects on intestinal morphology. Again, our present study confirmed the above research results. Also, our present study provided the first evidence that resveratrol could improve the relative jejunum weight and the relative ileum length of broilers under HS. The above results indicate that resveratrol can protect the intestinal morphology of broilers under HS. However, the specific protective mechanism of resveratrol on the intestine remains to be further explored.

Studies have shown that oxidative stress induced by HS leads to oxidative damage to the intestinal mucosa of broilers and disrupts the redox balance of the animal body, resulting in the production of a large amount of reactive oxygen species (ROS) and the reduction of antioxidant capacity [1,28]. It is known that excessive ROS induced by HS could cause intestinal mucosal epithelial cell apoptosis, which affects the absorption function of the intestine, disturbs the permeability of the intestinal wall, and impairs the intestinal barrier [5,6]. However, tissues and cells can eliminate ROS through antioxidant enzymes (CAT, GPX, SOD) and exert antioxidant effects [29]. SOD is the body’s first line of defense against free radicals, and GPX can scavenge superoxide anion free radicals, both of which play a crucial role in the body’s oxidation and antioxidant balance [30]. GST is an important detoxification enzyme, which uses the reduction effect of glutathione (GSH) to remove the active oxygen free radicals produced in the cell and repair oxidative damage. This present study showed that GPX, GST, and SOD activities in the jejunum mucosa of broilers under HS were substantially lower, indicating that oxidative stress occurred in the jejunum. This result was further substantiated by the elevated concentration of PC in the jejunum mucosa of broilers, which may be due to the considerable accumulation of ROS in the intestine. Li et al. [31] found that HS treatment of jejunal epithelial cells at 43 °C could significantly reduce the activity of antioxidant enzymes such as SOD and GPX in jejunal epithelial cells and increase the content of MDA. Studies have also shown that resveratrol can be used as an effective plant antitoxin polyphenol to activate the body’s antioxidant enzyme system and regulate antioxidant-related signal pathways to exert its antioxidant function, ensuring the intestinal health and safe growth of animals [32]. In vitro experiments showed that resveratrol improved SOD-1, CAT, and GPX activities in small intestinal epithelial cells. Therefore, resveratrol can be used as an effective plant antitoxin polyphenol compound to relieve heat stress-induced antioxidant damage in broilers.

The Nrf2 signaling pathway is an essential anti-oxidative stress mechanism in the body [33], which plays a vital role in the process of anti-oxidative stress of tissues and organs such as the liver, brain, kidney, and intestine. Nrf2 acts as a transcription factor to regulate the expression of antioxidant-related genes which are essential for maintaining the organism redox balance, such as *SOD*, *GPX*, *HO-1*, and *CAT* [19,20], while *Keap1* has a critical regulatory function on the transcriptional activity of Nrf2 [34]. Another compelling result of the present study is that HS inhibits the activation of the Nrf2 pathway. This inhibition was reflected in the upregulation of Keap1 and the downregulation of *Nrf2*, *SOD1*, and *GPX*. A previous study has shown that in addition to its role as a reducing agent, resveratrol can also play an antioxidant role by activating the antioxidant signaling pathway in cells or organisms [35]. Wang et al. [36] showed that resveratrol protected the intestinal barrier by increasing *HO-1* gene expression; this effect was mainly achieved by activating Nrf2 and then up-regulating *HO-1* expression. Moreover, Cheng et al. [37] showed that the HT-induced decrease of *Nrf2*, *SOD1*, and *GPX* mRNA expression levels in the rat jejunum was alleviated by dietary resveratrol supplementation, which showed that resveratrol could activate the antioxidant defense mechanism through the Nrf2 signaling pathway and maintain the body’s health. Excitingly, the present study found that dietary resveratrol supplementation not only increased the activities of *GPX* and *GST* and the mRNA level for *Nrf2* and *SOD1* but also decreased the content of PC and the mRNA level *Keap1* in the jejunum mucosa of broilers under HS. The above results suggested that dietary resveratrol supplementation is an effective means to prevent the disorder of the antioxidant system induced by HS, and the mechanism of action may be attributed to the activation of the Nrf2 signaling pathway.

## 5. Conclusions

In conclusion, the results of this study indicated that HS leads to intestinal oxidative stress. This stress adversely affected the intestine’s growth and development and ultimately decreased the growth performance of broilers. Dietary resveratrol could improve broiler’s growth under HS by relieving HS’s adverse effects on intestinal morphology and improving the intestinal mucosa’s antioxidant capacity. The mechanism of action may be related to the activation of the Nrf2 signaling pathway induced by resveratrol. However, further investigations will be required to elucidate the exact mechanism of action.

## Figures and Tables

**Table 1 animals-11-01427-t001:** Composition and nutrient levels of the basal diets.

Ingredients		Calculated Nutritional Levels	
Corn	57.30	Metabolizable energy (Kcal/kg)	3044
Soybean meal	35.20	Crude protein (%)	20.02
Soybean oil	3.99	Methionine (%)	0.40
Limestone	1.12	Ca (%)	0.90
CaHPO_4_·2H_2_O	1.61	Available phosphorus (%)	0.40
Methionine	0.10	Lysine (%)	1.07
Salt	0.30	Methionine + Cysteine (%)	0.73
Choline chloride	0.15		
Premix ^1^	0.23		
Total	100.00		

^1^ The vitamin mix provided (per kg of complete diet): vitamin A, 4000 IU; vitamin D_3_, 800 IU; vitamin E, 44 IU; vitamin K_3_, 0.5 mg; thiamine,1 mg; riboflavin, 3.75 mg; vitamin B6,1 mg; vitamin B_12_,15 μg; niacin, 10 mg; biotin, 0.2 mg; pantothenic acid, 12 mg; folic acid, 1.3mg; Cu, 10 mg as CuSO_4_·5H_2_O; Fe, 80 mg as FeSO4; I, 0.6mg as KI; Zn, 100 mg as ZnSO_4_; Mn, 25 mg as MnSO_4_; Se, 0.15 mg as Na_2_SeO_3_.

**Table 2 animals-11-01427-t002:** Sequences of the primers used for the detection of gene expression levels.

Gene	F (5′-3′)	R (5′-3′)	Accession No.
*Nrf2*	TTCGCAGAGCACAGATACTTC	TGGGTGGCTGAGTTTGATTAG	NM_205117.1
*HO-1*	TGTCCCTCCACGAGTTCAAG	CTCCAGTTGCTGCCATAGAA	NM_205344.1
*NQO1*	CTCCGAGTGCTTTGTCTACGA	ATGGCTGGCATCTCAAACC	NM_001277621.1
*Keap1*	CTGCTGGAGTTCGCCTACAC	CACGCTGTCGATCTGGTACA	KU321503.1
*GST*	GGAAGCCATTTTAATGACAGA	TCCTTTAAAAGCCTGTAGCAGA	XM_015284825.2
*GPX*	ACGGCGCATCTTCCAAAG	TGTTCCCCCAACCATTTCTC	NM_001277853.2
*SOD1*	AGGTCCAGCATTTCCAGTTAG	GGCGTATGACCCTAGCAACA	NM_205064.1
*CAT*	GGCGTATGACCCTAGCAACA	TCTGATAATTGGCCACGCGA	NM_001031215.2
*β-actin*	TGATATTGCTGCGCTCGTTG	AACCATCACACCCTGATGTCTG	NM_205518.1

Nrf2 = NF-E2-related factor 2; HO-1 = Heme oxygenase 1; NQO1 = NAD (P)H/quinone oxidoreductase 1; Keap1 = Kelch-like ECH-associated protein 1; GST = glutathione S-transferase; GPX = Glutathione peroxidase; SOD1 = Superoxide dismutase 1; CAT = Catalase.

**Table 3 animals-11-01427-t003:** Effect of resveratrol on the growth performance of broilers under HS.

Items	CON	HS	HS + Resveratrol
Final BW (g)	2324.49 ± 33.23 ^a^	2133.28 ± 35.89 ^b^	2267.21 ± 43.51 ^a^
ADG (g)	68.97 ± 1.72 ^a^	60.45 ± 1.49 ^b^	66.27 ± 0.56 ^a^
ADFI (g)	128.53 ± 1.50	124.71 ± 2.99	125.08 ± 1.30
F/G	1.85 ± 0.04	2.06 ± 0.09	1.90 ± 0.03

CON: basal diet; HS: heat stress + basal diet; HS + Resveratrol: heat stress + basal diet with 400 mg/kg resveratrol; *n* = 6. ^a,b^ Different superscripts within a row indicate a significant difference (*p* < 0.05).

**Table 4 animals-11-01427-t004:** Effect of resveratrol on intestinal characteristics of broilers under HS.

Items	CON	HS	HS + Resveratrol
Relative intestinal weight (g/kg of live BW)			
Duodenum (g/kg)	4.72 ± 0.13	4.25 ± 0.17	4.58 ± 0.20
Ileum (g/kg)	7.54 ± 0.18	6.85 ± 0.29	7.20 ± 0.42
Jejunum (g/kg)	10.41 ± 0.19 ^a^	9.11 ± 0.47 ^b^	10.28 ± 0.50 ^a^
Relative intestinal length (cm/kg of live BW)			
Duodenum (cm/kg)	13.23 ± 0.22	12.42 ± 0.42	12.98 ± 0.24
Ileum (cm/kg)	32.23 ± 0.78 ^a^	28.61 ± 0.78 ^b^	31.25 ± 0.59 ^a^
Jejunum (cm/kg)	31.62 ± 0.23 ^a^	28.97 ± 0.55 ^b^	30.73 ± 0.28 ^a,b^

CON: basal diet; HS: heat stress + basal diet; HS + Resveratrol: heat stress + basal diet with 400 mg/kg resveratrol; *n* = 12. ^a,b^ Different superscripts within a row indicate a significant difference (*p* < 0.05).

**Table 5 animals-11-01427-t005:** Effect of resveratrol on the morphology of jejunum of broilers under HS.

Items	CON	HS	HS + Resveratrol
Villus height (μm)	1542.56 ± 54.65 ^a^	1334.31 ± 51.87 ^b^	1523.67 ± 35.91 ^a^
Crypt depth (μm)	247.8 ± 9.36	253.27 ± 6.86	248.88 ± 6.03
Villus height/crypt depth	6.23 ± 0.26 ^a^	5.25 ± 0.28 ^b^	6.10 ± 0.31 ^a,b^

CON: basal diet; HS: heat stress + basal diet; HS + Resveratrol: heat stress + basal diet with 400 mg/kg resveratrol; *n* = 12. ^a,b^ Different superscripts within a row indicate a significant difference (*p* < 0.05).

**Table 6 animals-11-01427-t006:** Effect of resveratrol on antioxidant status in jejunum of broilers under HS.

Items	CON	HS	HS + Resveratrol
CAT (U/mg prot)	227.09 ± 12.29	200.39 ± 7.69	224.70 ± 10.12
GPX (U/mg prot)	375.44 ± 11.06 ^a^	291.49 ± 8.24 ^b^	352.37 ± 5.53 ^a^
GR (U/mg prot)	4.88 ± 0.65	4.24 ± 0.28	4.51 ± 0.35
GST (U/mg prot)	733.36 ± 10.82 ^a^	663.81 ± 14.03 ^b^	720.82 ± 11.98 ^a^
SOD (U/mg prot)	18.80 ± 1.23 ^a^	14.50 ± 0.88 ^b^	16.38 ± 0.90 ^a,b^
MDA (nmol/mg prot)	1.87 ± 0.29	2.40 ± 0.32	1.90 ± 0.28
PC (nmol/mg prot)	6.40 ± 0.46 ^b^	9.75 ± 1.03 ^a^	7.46 ± 0.82 ^b^

CON: basal diet; HS: heat stress + basal diet; HS + Resveratrol: heat stress + basal diet with 400 mg/kg resveratrol; *n* = 12. ^a,b^ Different superscripts within a row indicate a significant difference (*p* < 0.05).

**Table 7 animals-11-01427-t007:** Effects of resveratrol on the mRNA levels of Nrf2 signaling pathway-related genes in jejunum of broilers under HS.

Items	CON	HS	HS + Resveratrol
*Nrf2*	1.00 ± 0.05 ^a^	0.67 ± 0.02 ^b^	0.97 ± 0.03 ^a^
*Keap1*	1.00 ± 0.05 ^b^	1.44 ± 0.01 ^a^	1.07 ± 0.08 ^b^
*HO-1*	1.00 ± 0.06	0.83 ± 0.03	0.88 ± 0.05
*NQO1*	1.00 ± 0.05	0.88 ± 0.14	0.95 ± 0.15
*CAT*	1.00 ± 0.11	0.88 ± 0.10	0.92 ± 0.08
*SOD1*	1.00 ± 0.05 ^a^	0.62 ± 0.26 ^b^	0.93 ± 0.12 ^a^
*GST*	1.00 ± 0.07	0.83 ± 0.04	0.90 ± 0.07
*GPX*	1.00 ± 0.06 ^a^	0.71 ± 0.10 ^b^	0.94 ± 0.08 ^a,b^

CON: basal diet; HS: heat stress + basal diet; HS + Resveratrol: heat stress + basal diet with 400 mg/kg resveratrol; *n* = 12. ^a,b^ Different superscripts within a row indicate a significant difference (*p* < 0.05).

## Data Availability

The data presented in this study are available on request from the corresponding author.

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
