# Peer review of "Effects of Resveratrol on Growth Performance, Intestinal Development, and Antioxidant Status of Broilers under Heat Stress"

_animals, 2021, doi:10.3390/ani11051427_

Round 1
Reviewer 1 Report
What does it mean “AA broilers”? what kind of hybrid it is? explain, please
If diet was presented in the previous work (line 68), probably table 1 (line 76) is redundant? consider, please
Table 3 should be placed after 3.1 paragraph (about line 131)
Table 4 – it seems that the most important effect in this case was the influence of resveratrol on the increase of intestine weight. There were not differences between CON and HS+ resveratrol. throughout the manuscript, more emphasis should be placed on comparing the HS+ resveratrol group with the control group than with the HS group. Such a comparison indicates the protective effect of resveratrol in terms of the heat stress.
Table 6 should be placed after 3.4 paragraph (about line 156)
The minor revision is recommended.
Author Response
Comment 1: What does it mean “AA broilers”? what kind of hybrid it is? explain, please
Response: AA broilers is the abbreviation for Arbor Acres broilers, which is one of the oldest and most respected names in the poultry industry. In the revised manuscript, we have changed “AA” to “Arbor Acres”. (Line 71)
Comment 2: If diet was presented in the previous work (line 68), probably table 1 (line 76) is redundant? consider, please
Response: There’s a mistake, we are very sorry for our negligence. In the revised manuscript, we have changed “Diets were consistent with our previous reports [3,5],” to “Diet was formulated according to our previous reports[21]”. (Line 70)
Comment 3: Table 3 should be placed after 3.1 paragraph (about line 131)
Response: Thanks for your good suggestion. Table 3 has been placed after 3.1 paragraph. (Lines 136-138)
Comment 4: Table 4 – it seems that the most important effect in this case was the influence of resveratrol on the increase of intestine weight. There were not differences between CON and HS+ resveratrol throughout the manuscript, more emphasis should be placed on comparing the HS+ resveratrol group with the control group than with the HS group. Such a comparison indicates the protective effect of resveratrol in terms of the heat stress.
Response: Although the data are different (raised or lowered) between CON and HS+ resveratrol groups, there were no statistically significant differences between CON and HS+ resveratrol throughout the manuscript, indicating that resveratrol was effective in ameliorating the adverse effects of heat stress. We have added “There were no statistically significant differences between CON and HS+ resveratrol groups” to the revised manuscript. (Lines 134,143,151,160,168)
Comment 5: Table 6 should be placed after 3.4 paragraph (about line 156)
Response: Thanks for your good suggestion. Table 6 has been placed after 3.4 paragraph. (Lines 161-163)
Reviewer 2 Report
Dear authors. This is an interesting and well-written manuscript. I have only a few minor suggestions for your revision.
- L68 - specify "AA" broilers
- L69 - please also give standard deviation of the mean
- S2.1 - please provide more information on the stable conditions, e.g., distribution of radiators and distribution of treatment pens
- L96 - should read "2,500 x g"
- Tab1 - did you analyse this data? if so, name the used methods in M&M section; specify "available phosphorus"
- L186-187 - please rewrite - is "damaged" the right word?
- L209 - should read "found", not "founded"
- L218 - "anti-oxidative stress damage", please explain
Author Response
Comment 1: L68 - specify "AA" broilers
Response: AA broilers is the abbreviation for Arbor Acres broilers, which is one of the oldest and most respected names in the poultry industry. In the revised manuscript, we have changed “AA” to “Arbor Acres”. (Line71)
Comment 2: L69 - please also give standard deviation of the mean
Response: The standard deviation of the mean is 20, which was added to in the revised manuscript. (Line 71)
Comment 3: S2.1 - please provide more information on the stable conditions, e.g., distribution of radiators and distribution of treatment pens
Response: We have added “The temperature and the relative humidity were automatically controlled by the environmental control systems (Big Herdsman, Qingdao, China)” to the revised manuscript. (Lines 77-78)
Comment 4: L96 - should read "2,500 x g"
Response: The manuscript has been revised as required. (Line 99)
Comment 5:Tab1 - did you analyse this data? if so, name the used methods in M&M section; specify "available phosphorus"
Response: Nutritional levels in Table 1 were calculated nutritional levels. We have added “calculated” to Table 1. (Line 121)
Comment 6: L186-187 - please rewrite - is "damaged" the right word?
Response: In the revised manuscript we changed "damaged" to "was harmful to". (Line 192)
Comment 7: L209 - should read "found", not "founded"
Response: we are very sorry for our negligence, we have changed “founded” to “found”. (Line 215)
Comment 8: L218 - "anti-oxidative stress damage", please explain
Response: In the revised manuscript, we have changed “anti-oxidative stress damage” to “the process of anti-oxidative stress”. (Line 224)
Reviewer 3 Report
The paper is well written and the data are clearly presented.
It has clarified a part of the mode of action.
The only major remark is that ue to the experimental design the authors conclude that resveratrol alliviates the negative effects of Heat stress. however the experiment would have been better controlled if the resveratrol treatment was also given to the control in a 2 x 2 design. Based on this study no conclusions can be made with temeratures in the thermoneutral zone. If literature is available it should be mentioned in a few sentences.
Minor remarks:
line 48: mention some references. Later you do. more important is that also be other nutritional intervantions, like low protein, vit.C, etc, heat stress can also be reduced.
line 68:
refer to table 1 and not to previous work.
table 3: normally a reduction in feed intake is observed during heat stressand a lower gain a consequence. . In this experiment the feed intake in the HS treatment was not much reduced. Sice the authors mention similar studies in the group, I would have expected this resukt would be discussed and compared with previous experiments.
table 4 would have added value if also the absolute weight of the different segments of the intestines were presented.
Author Response
Comment 1: The only major remark is that ue to the experimental design the authors conclude that resveratrol alliviates the negative effects of Heat stress. however the experiment would have been better controlled if the resveratrol treatment was also given to the control in a 2 x 2 design. Based on this study no conclusions can be made with temeratures in the thermoneutral zone. If literature is available it should be mentioned in a few sentences.
Response: We have added “Our previous study found that resveratrol could improve the muscle antioxidant function of broilers reared under nor-mal ambient temperature [18]” in the revised manuscript. (Lines 54-55)
Comment 2: line 48: mention some references. Later you do. more important is that also be other nutritional intervantions, like low protein, vit.C, etc, heat stress can also be reduced.
Response: We have added "….such as vitamin E and vitamin C [11], glutamine [12]" in the revised manuscript. (Lines 48-49)
Comment 3: line 68: refer to table 1 and not to previous work.
Response: We have have changed “Diets were consistent with our previous reports [3,5]” to “Diet was formulated according to our previous reports [21]. (Lines 70)
Comment 4: table 3: normally a reduction in feed intake is observed during heat stress and a lower gain a consequence. In this experiment the feed intake in the HS treatment was not much reduced. Since the authors mention similar studies in the group, I would have expected this result would be discussed and compared with previous experiments.
Response: We have added “However, we haven’t found significant difference in ADFI, which may be related to the lower heat treated temperature (33±1 ℃) of present study compared to the study of Liu et al. (37±2 ℃) [14]” in the revised manuscript. (Lines 177-179)
Comment 5: table 4 would have added value if also the absolute weight of the different segments of the intestines were presented.
Response: Because the final BW of broiler was significantly affected, the relative intestinal weight is more valuable than the absolute weight. Therefore, the absolute weight of intestine was not presented in table 4.